# OCCPROPHET: PUSHING EFFICIENCY FRONTIER OF CAMERA-ONLY 4D OCCUPANCY FORECASTING WITH OBSERVER-FORECASTER-REFINER FRAMEWORK

**Junliang Chen**[*], **Huaiyuan Xu**[*], **Yi Wang**[†], **Lap-Pui Chau**[†]
The Hong Kong Polytechnic University
jun-liang.chen@connect.polyu.hk
{huaiyuan.xu, yi-eie.wang, lap-pui.chau}@polyu.edu.hk

## ABSTRACT

Predicting variations in complex traffic environments is crucial for the safety of autonomous driving. Recent advancements in occupancy forecasting have enabled forecasting future 3D occupied status in driving environments by observing historical 2D images. However, high computational demands make occupancy forecasting less efficient during training and inference stages, hindering its feasibility for deployment on edge agents. In this paper, we propose a novel framework, *i.e.*, OccProphet, to efficiently and effectively learn occupancy forecasting with significantly lower computational requirements while improving forecasting accuracy. OccProphet comprises three lightweight components: Observer, Forecaster, and Refiner. The Observer extracts spatio-temporal features from 3D multi-frame voxels using the proposed Efficient 4D Aggregation with Tripling-Attention Fusion, while the Forecaster and Refiner conditionally predict and refine future occupancy inferences. Experimental results on nuScenes, Lyft-Level5, and nuScenes-Occupancy datasets demonstrate that OccProphet is both training-and inference-friendly. OccProphet reduces 58%∼78% of the computational cost with a 2.6× speedup compared with the state-of-the-art Cam4DOcc. Moreover, it achieves 4%∼18% relatively higher forecasting accuracy. Code and models are publicly available at https://github.com/JLChen-C/OccProphet.

## 1 INTRODUCTION

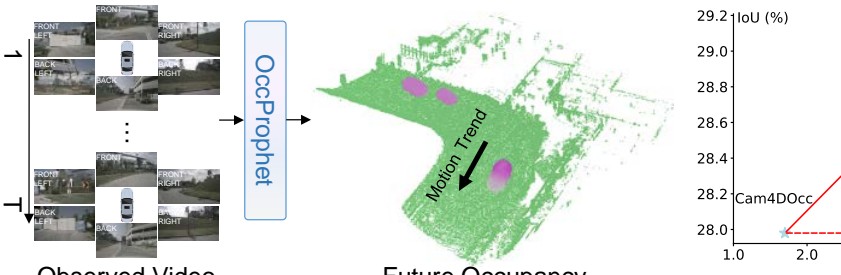

Figure 1: Illustration of OccProphet. OccProphet only receives multi-camera video input and produces future occupancies.

Figure 2: Comparison of performance between Cam4DOcc and OccProphet.

Autonomous driving holds significant promise for reshaping transportation and urban mobility. Perceiving 3D surroundings is critical for autonomous vehicles. There are typically two 3D perception paradigms: detection-based perception and occupancy-based perception. The detection-based paradigm, such as monocular or multi-view 3D object detection (Chen et al., 2016; Wang et al., 2021; Huang et al., 2021; Wang et al., 2022), equips autonomous vehicles with 3D perception capabilities by detecting traffic participants and assigning them 3D bounding boxes. However, due to

---

[*] Equal contribution    [†] Corresponding author

limitations in pre-defined object categories and rigid detection boxes, the detection-based paradigm struggles to generalize to unknown objects and irregular structures, which are likely to appear in real-world traffic scenarios. To relieve these constraints, the occupancy-based perception paradigm (Huang et al., 2023a; Wei et al., 2023b; Wang et al., 2023b; Tian et al., 2023; Xu et al., 2025) offers a more general and fine-grained representation of the environment through learning occupied states in 3D space. This paradigm provides a stronger perception ability to autonomous vehicles, enabling them to better comprehend complex traffic environments.

Despite advancements in occupancy perception, most existing methods can only perceive the past and present states of the environment. They lack the ability to capture and understand environmental dynamics and subsequently forecast the future scene. Forecasting is essential for safe planning, which assists autonomous vehicles to avoid potential collisions (Ding et al., 2019; Song et al., 2020; Ding et al., 2021; Huang et al., 2023b; Pan et al., 2024). Although some bird's-eye view (BEV) approaches have achieved object motion forecasting in the environment(Hu et al., 2021; Zhang et al., 2022; Fang et al., 2023; Ferenczi et al., 2024), their forecasts are restricted on a 2D plane. This 2D forecasting limits the comprehensive understanding of the entire 3D dynamic scene. In light of this limitation, occupancy forecasting methods (Weng et al., 2021; 2022; Mersch et al., 2022; Khurana et al., 2023; Agro et al., 2024) shift to predict future 3D occupancy for the whole environment. However, these methods rely on point cloud inputs from expensive LiDAR kits. To explore more cost-effective solutions, Cam4DOcc (Ma et al., 2024a) introduces a camera-only benchmark and baseline for occupancy forecasting.

While Cam4DOcc achieves remarkable performance in occupancy forecasting compared to its counterparts, the high computational cost makes it less efficient during training and inference. It hampers feasibility of deployment on edge agents, such as autonomous vehicles that operate under restricted computational budgets. In this paper, we propose a novel framework, dubbed as *OccProphet* (shown in Figure 1), to efficiently and effectively perform camera-only occupancy forecasting. In OccProphet, we design three lightweight components to forecast future states: the *Observer*, *Forecaster*, and *Refiner*. The Observer adopts 4D feature aggregation and a tripling-attention fusion strategy on the reduced-resolution features to extract spatio-temporal information efficiently. The Forecaster then infers future states according to the scene condition and the Observer's outputs. Finally, the Refiner enhances the quality of the forecast results through spatio-temporal interactions. The main contributions of this paper are summarized as follows:

- We propose OccProphet, a novel camera-only occupancy forecasting framework, which is both efficient and effective during training and inference, towards on-vehicle deployment.

- We design a lightweight Observer-Forecaster-Refiner pipeline for OccProphet. The Observer extracts spatio-temporal features from historical observations; the Forecaster conditionally predicts coarse future states; the Refiner promotes forecasting accuracy.

- Experimental results demonstrate that OccProphet achieves higher forecasting accuracy with less than half the computational cost of Cam4DOcc. These improvements are consistently observed across the nuScenes (Caesar et al., 2020), Lyft-Level5 (Houston et al., 2021), and nuScenes-Occupancy (Wang et al., 2023a) datasets, highlighting the superior efficiency and effectiveness of OccProphet (shown in Figure 2).

## 2 RELATED WORK

### 2.1 OCCUPANCY PREDICTION

Occupancy prediction aims at modeling the current 3D occupancy layout in space, by observing historical and current environments. Occupancy prediction is adept at providing 3D dense descriptions for complex traffic scenarios, thereby garnering increasing attention from academia and industry. SSCNet (Song et al., 2017) was the first semantic occupancy prediction work, which simultaneously predicted occupied voxels and their semantics for an indoor scene using a depth image. MonoScene (Cao & De Charette, 2022) extends SSCNet to outdoor scenarios by using an RGB image and incorporating stronger supervisions. Training and evaluating occupancy prediction networks require benchmarks with ground truth occupancy labels, which are challenging due to the complexity of densely annotating 3D outdoor driving scenes. Wang *et al.* (Wang et al., 2023b) propose OpenOccupancy, the first large-scale benchmark for semantic occupancy prediction, which covers multiple sensing modalities and provides high-resolution dense occupancy annotations. Occ3d, developed by

Tian *et al.* (Tian et al., 2023), is another widely used benchmark for occupancy prediction, whose high-quality labels are from the proposed technique of image-guided occupancy label refinement.

Considering the inherently dense nature of depicting 3D environments, ideal occupancy perception emphasizes balancing efficiency and accuracy. Although some studies explore sparse queries (Li et al., 2023b) or tri-perspective view (TPV) representation (Huang et al., 2023a) to elevate the efficiency of occupancy prediction, they inevitably sacrifice fine-grained details of 3D space. In contrast, many other methods (Zhang et al., 2023; Wei et al., 2023b; Ma et al., 2024b) utilize 3D feature volumes to preserve 3D details of the scene, leading to higher occupancy prediction accuracy. Recently, COTR (Ma et al., 2024b) proposes a compact occupancy representation that preserves geometric details while reducing computational costs. Despite the significant advancements in occupancy prediction, including comprehensive benchmarks and powerful algorithms, all existing methods focus exclusively on current occupancy and overlook future occupancy, which reflects potential variations in the 3D environment.

## 2.2 OCCUPANCY FORECASTING

Occupancy forecasting targets to predict future occupancy, starting from the current timestamp. Previous dominant works mainly adopt the BEV perspective for occupancy forecasting, reasoning about 2D occupancy changes on a BEV plane. For example, FIERY (Hu et al., 2021) extracts BEV features from multi-view image inputs and utilizes a temporal model with 3D convolution to capture spatio-temporal states, which are then used to recursively forecast future instances states. BEVerse (Zhang et al., 2022) introduces a unified BEV representation framework that jointly achieves object perception and occupancy forecasting using multi-task supervision. To better align spatio-temporal information, TBP-Former (Fang et al., 2023) designs a pose-synchronized BEV encoder to synchronize multi-frame BEV features during occupancy forecasting. While these BEV-based methods deliver impressive performance for forecasting pre-defined semantic categories (*e.g.*, vehicles), they struggle to (1) forecast the motion of out-of-distribution objects and (2) capture height information in the environment. In contrast, our method forecasts class-agnostic occupancy from a 3D perspective, rather than BEV, thus enabling autonomous vehicles to monitor, comprehend, and reason about 3D dynamics in the physical world.

To address the limitations of BEV occupancy forecasting, researchers have recently shifted their focus toward forecasting 3D occupancy without considering semantics. Specifically, Khurana *et al.* (Khurana et al., 2023) treat LiDAR point cloud forecasting as a proxy task for the occupancy forecasting task, where point cloud rendering is used to bridge the two tasks. UnO (Agro et al., 2024) takes LiDAR point clouds as input and performs occupancy forecasting using the proposed unsupervised learning paradigm, in which the forecasted occupancy should align with pseudo occupancy labels generated from future LiDAR data. However, these methods rely on point clouds from expensive LiDAR kits, leading to increased costs when implemented in autonomous vehicles.

Compared to LiDAR-based approaches, camera-only occupancy forecasting offers a promising alternative with significantly lower costs. Cam4DOcc (Ma et al., 2024a) introduces a comprehensive benchmark and dataset to evaluate camera-only occupancy forecasting algorithms on both movable and static objects beyond pre-defined categories. Additionally, it proposes a strong camera-only baseline for occupancy forecasting. However, this approach is still far from real-world application due to its high computational demands. Drive-OccWorld (Yang et al., 2024) introduces extra action condition inputs and planning supervision to enhance performance. In this paper, we propose a novel end-to-end framework with faster speed, and higher accuracy, enabling efficient and effective occupancy forecasting in a pure camera-only setting.

## 3 OCCPROPHET

### 3.1 OVERVIEW

The overview of the proposed OccProphet is illustrated in Figure 3. Given multi-frame surround-view RGB images as input, 2D features are extracted using a shared image encoder. These 2D features are subsequently lifted into 3D space and aggregated into multi-frame 3D voxel features through depth estimation and voxel pooling. We design the following unshared pipeline for each of the occupancy and occupancy flow branches. Specifically, the pipeline consists of four components: the Observer, Forecaster, Refiner, and Predictor. The Observer module efficiently and ef-

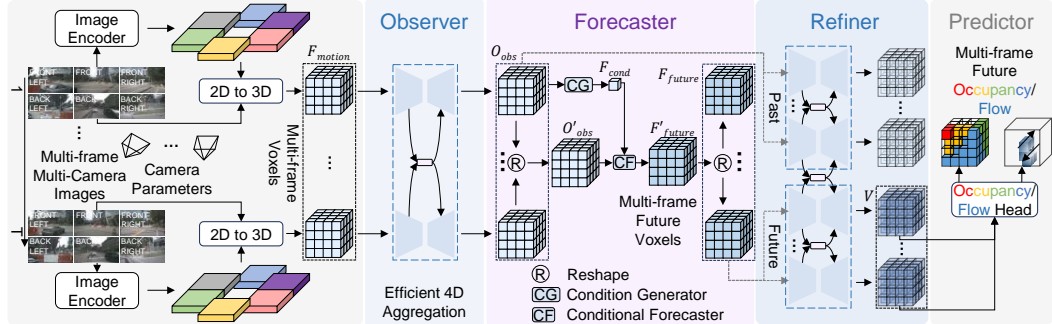

Figure 3: Overview of OccProphet. It receives multi-frame images from surround-view cameras as input and outputs future occupancy or occupancy flow. It consists of four key components: the Observer, Forecaster, Refiner, and Predictor. The Observer module aggregates spatio-temporal information. The Forecaster module conditionally generates preliminary representations of future scenarios. These preliminary representations are refined by the Refiner module. Finally, the Predictor module produces the final predictions of future occupancy or occupancy flow.

fectively aggregates spatio-temporal information within multi-frame observations (*i.e.*, multi-frame 3D voxel features). The Observer's output then undergoes a Forecaster, which adaptively predicts future states, ensuring flexibility across diverse traffic conditions. The Refiner module further enhances the quality of these predictions by enabling cross-frame interactions. Finally, the Predictor module decodes the refined future states into either occupancy or occupancy flow.

## 3.2 OBSERVER

The Observer takes the 4D motion-aware feature $F_{motion}$ as input, and generates a spacetime-aware representation. Let $\mathcal{I}_t = \{I_t^v\}_{v=1}^{N_{cam}}$ denote the multi-camera RGB images at timestamp $t \in \{1, \ldots, T\}$, where $N_{cam}$ refers to the total number of surround-view cameras, and $T$ is the total number of input frames. A shared image encoder (*i.e.*, ResNet (He et al., 2016)) is applied to $\{\mathcal{I}_t | t = 1, \ldots, T\}$ to extract 2D features. For each frame, these 2D features are projected to 3D space and then aggregated into voxelized 3D features (Philion & Fidler, 2020). The 3D features from multiple frames are aligned into the current-frame coordinate system using 6 degrees of freedom (6-DoF) ego-vehicle poses. We then concatenate the aligned 3D features into a 4D feature $F \in \mathbb{R}^{T \times C \times X \times Y \times Z}$, where $C$ is the number of channels, and $(X, Y, Z)$ represents the size of voxelized 3D feature volume. Subsequently, the motion-aware 4D feature $F_{motion} \in \mathbb{R}^{T \times (C+6) \times X \times Y \times Z}$ is generated by concatenating 6-DoF ego-vehicle poses.

To generate the spacetime-aware representation, directly processing the 4D motion-aware feature using convolutional operations is intuitive. However, the direct processing imposes a substantial computational burden, and ignores the fact that a large portion of the 3D space is unoccupied, which leads to the inherent sparsity of the motion-aware feature. To address this issue, we efficiently and effectively generate the spacetime-aware feature from $F_{motion}$ using the Observer module. The Observer comprises an Efficient 4D Aggregation module and a Tripling-Attention Fusion module.

### 3.2.1 EFFICIENT 4D AGGREGATION

Directly aggregating the original 4D feature $F_{motion}$ will incur a high computational cost. For efficiency, we design the Efficient 4D Aggregation (E4A) module to first produce compact features through downsampling, and then exploit spatio-temporal interactions on the compact features to achieve aggregation, followed by the upsampling process to compensate for the information loss. The architecture of the E4A module is shown in Figure 4. We first reduce the channel number of $F_{motion}$ from $(C + 6)$ to $C$ through a 3D convolution, thereby forming a feature $O \in \mathbb{R}^{T \times C \times X \times Y \times Z}$. We gradually downsample $O$

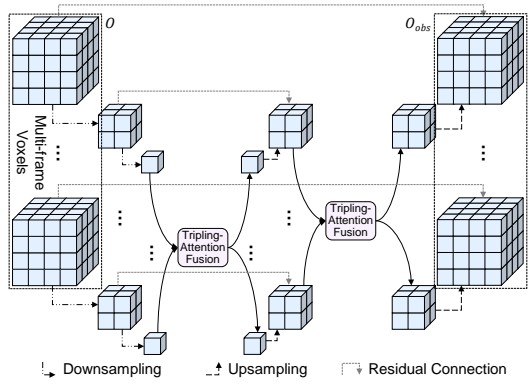

Figure 4: Efficient 4D Aggregation (E4A).

to reduce computation during aggregation. However, such downsampling inevitably results in non-negligible information loss, especially for small objects. To compensate for this loss, we conduct two operations. On one hand, we perform spatio-temporal interactions on the downsampled features, that is, the Tripling-Attention Fusion module (to be illustrated in Section 3.2.2). On the other hand, the post-interaction features are upsampled, and further summed with the features at the same resolution prior to downsampling. These two operations continue until the resolution of the upsampled feature matches the resolution of the initial motion-aware feature.

The output 4D representation of the E4A module possesses spatio-temporal awareness. It effectively preserves the 3D geometric details of the environment–details that the BEV and TPV representations are unable to capture. As shown in Table 1, using the E4A representation achieves higher performance than the BEV and TPV representations with fewer parameters and a slight increase in computational costs.

Table 1: Comparisons with different representation styles. $N_p$: Total number of parameters.

| Style | $N_p$ (M) | FLOPs (G) | $\tilde{\text{IoU}}_f$ (%) |
|---|---|---|---|
| BEV | 102 | 1370 | 26.50 |
| TPV | 181 | 1574 | 26.96 |
| E4A | 67 | 1737 | **27.50** |

### 3.2.2 TRIPLING-ATTENTION FUSION

A 4D feature can be considered as a combination of multiple 3D voxel-wise features along the temporal dimension. The Tripling-Attention Fusion (TAF, shown in Figure 5) module is specifically designed to facilitate spatio-temporal interactions across multiple 3D features, as depicted on the left of Figure 5. Notably, downsampling reduces computational cost by lowering feature resolution, while the Tripling-Attention Fusion module makes a further step through the proposed tripling operation, as illustrated on the right of Figure 5.

The tripling operation is designed to understand the 3D space from three complementary and compact perspectives, which can retain 3D scene information with fewer computational costs. Specifically, a tripling operation decomposes a 3D feature into three distinct branches: scene, height, and BEV. This decomposition compresses 3D features into 1D and 2D features, significantly reducing the subsequent computa-

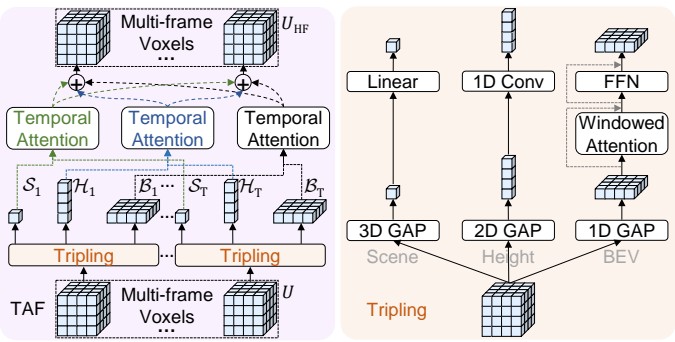

Figure 5: Tripling-Attention Fusion (left) and Tripling (right).

tional overhead. The scene branch can extract the global context of the corresponding frame, providing an overall understanding of the scenario. The height branch retains vertical details, serving as complementary clues to the 2D BEV branch to enhance the representation capability of the 3D geometric information. The three branches can be computed as follows:

$$\mathcal{S} = \text{Act}(\text{Norm}(\text{Linear}(\text{GAP}_{3D}(U)))), \tag{1}$$

$$\mathcal{H} = \text{Act}(\text{Norm}(\text{Conv}_{1D}(\text{GAP}_{2D}(U)))), \tag{2}$$

$$\mathcal{B} = \text{W-MSA}(\text{GAP}_{1D}(U)), \tag{3}$$

where $U \in \mathbb{R}^{T \times C \times \frac{X}{2^i} \times \frac{Y}{2^i} \times \frac{Z}{2^i}}$ denotes the $i$-th downsampled feature input to the Tripling-Attention Fusion module. $\text{GAP}_{3D}$, $\text{GAP}_{2D}$, and $\text{GAP}_{1D}$ are 3D, 2D, and 1D global average pooling. Linear, $\text{Conv}_{1D}$, Norm, and Act refer to a single linear layer, 1D convolution, normalization, and activation, respectively. W-MSA is the window-based multi-head self-attention block (Liu et al., 2021). $\mathcal{S} \in \mathbb{R}^{T \times C \times 1 \times 1 \times 1}$, $\mathcal{H} \in \mathbb{R}^{T \times C \times 1 \times 1 \times \frac{Z}{2^i}}$, and $\mathcal{B} \in \mathbb{R}^{T \times C \times \frac{X}{2^i} \times \frac{Y}{2^i} \times 1}$ denote the outputs of scene, height, and BEV branches. After the tripling operation, we interact and fuse scene, height, and BEV features. Specifically, we first independently apply the temporal attention along the time axis to different branches, then sum over the three branches using the broadcast technique. The process is formulated as follows:

$$U_{\text{HF}} = \text{TA}_{\text{scene}}(\mathcal{S}) \oplus \text{TA}_{\text{height}}(\mathcal{H}) \oplus \text{TA}_{\text{BEV}}(\mathcal{B}), \tag{4}$$

where $\text{TA}_{\text{scene}}$, $\text{TA}_{\text{height}}$, and $\text{TA}_{\text{BEV}}$ denote temporal attention for scene, height, and BEV branches; $\oplus$ is broadcast-style plus; $U_{\text{HF}} \in \mathbb{R}^{T \times C \times \frac{X}{2^i} \times \frac{Y}{2^i} \times \frac{Z}{2^i}}$ is the output feature of the TAF module.

### 3.3 FORECASTER

Given a spatio-temporal representation $O_{\text{obs}} \in \mathbb{R}^{T \times C \times X \times Y \times Z}$ output from the Observer, the Forecaster (shown in Figure 6) is supposed to generate future states. We first reshape $O_{\text{obs}}$ by collapsing the time axis into the channel axis, resulting in the reshaped features $O'_{\text{obs}} \in \mathbb{R}^{TC \times X \times Y \times Z}$. A straightforward approach to forecasting is using a single linear layer to predict features for future frames. This approach encourages the network to learn static weights to fit different traffic scenarios. However, traffic situations vary significantly across different traffic scenarios, presenting distinct challenges in predicting future changes using a single linear layer. For instance, forecasting environmental changes is more difficult in a crowded intersection with many moving objects than on an empty highway with few vehicles. The spatio-temporal complexity in the former is far greater than in the latter.

Considering these challenges, we propose forecasting occupancy with flexibility to adapt to various traffic scenarios featuring diverse spatio-temporal complexities. To achieve this, we design a novel Forecaster module that predicts future states based on the overall environmental condition.

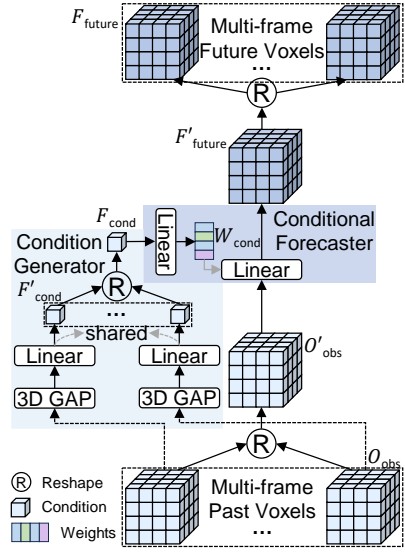

Figure 6: Architecture of Forecaster.

The Forecaster comprises a Condition Generator and a Conditional Forecaster. We first use a Condition Generator comprising a 3D GAP and a shared linear layer across different frames to extract the condition from the observation $O_{\text{obs}}$:

$$F'_{\text{cond}} = \text{Act}(\text{Norm}(\text{Linear}(\text{GAP}_{\text{3D}}(O_{\text{obs}})))), \qquad (5)$$

where $F'_{\text{cond}} \in \mathbb{R}^{T \times C \times 1 \times 1 \times 1}$ denotes the overall environmental condition, which is then reshaped into $F_{\text{cond}} \in \mathbb{R}^{TC}$ by collapsing the time axis into the channel axis. $F_{\text{cond}}$ is subsequently passed to a Conditional Forecaster to predict future states. Specifically, a linear layer is applied to $F_{\text{cond}}$ to produce adaptive weights for specific traffic scenarios. Another linear layer is then used to predict future states conditioned by these adaptive weights.

$$W_{cond} = \text{Linear}(F_{\text{cond}}), F'_{\text{future}} = \text{Linear}(O'_{\text{obs}}|W_{cond}), \qquad (6)$$

where $W_{cond} \in \mathbb{R}^{TC \times T'C}$ denotes the adaptive weights, $F'_{\text{future}} \in \mathbb{R}^{T'C \times X \times Y \times Z}$ is the future states of $T'$ future frames. $F'_{\text{future}}$ is reshaped into $F_{\text{future}} \in \mathbb{R}^{T' \times C \times X \times Y \times Z}$ to recover the time axis. $F_{\text{future}}$, as the preliminary feature for future environments, is then refined by the Refiner module.

### 3.4 REFINER

Since the Forecaster module predicts $F_{\text{future}}$ using linear projection, it inevitably lacks cross-frame interactions. The Refiner is designed to enhance the forecasted results via further interactions between future frames, as well as incorporating historical frames as supplementary information. The E4A module, described in Section Section 3.2.1, is a spatio-temporal interaction module. Furthermore, taking a review of the E4A, for any input feature $Q \in \mathbb{R}^{T \times C \times X \times Y \times Z}$, the function of the E4A module can be formulated as:

$$Q' = \text{E4A}(Q) = Q + \mathcal{F}(Q), \qquad (7)$$

where $Q' \in \mathbb{R}^{T \times C \times X \times Y \times Z}$ is the output feature of E4A, and $\mathcal{F}$ denotes the transformation function. Considering residual networks can aid in refinement and network optimization (He et al., 2016), it is reasonable to regard E4A as a *refinement transformation* for features, which also reduces the learning complexity of earlier modules. Building on this insight, we further introduce a Refiner reusing the E4A architecture to refine the forecasted future states. The Refiner is applied to the forecasted feature $F_{\text{future}}$ from the Forecaster and the feature output $O_{\text{obs}}$ from the Observer, producing an enhanced feature $V = E4A([O_{\text{obs}}, F_{\text{future}}])_{T+1:T+T',:,:,:,:} \in \mathbb{R}^{T' \times C \times X \times Y \times Z}$ for subsequent forecasting of occupancy and flow.

For fair comparisons, the Predictor for occupancy and occupancy flow prediction, and the overall training loss function follow those in Cam4DOcc (Ma et al., 2024a).

## 4 EXPERIMENTS

### 4.1 EXPERIMENTAL SETUPS

**Datasets.** Following Cam4DOcc, 700 out of 850 scenes in nuScenes (Caesar et al., 2020) and nuScenes-Occupancy (Wang et al., 2023a) datasets, and 130 out of 180 scenes in Lyft-Level5 (Houston et al., 2021) with occupancy labels are used for training, while the remaining scenes are used for evaluation. For nuScenes and nuScenes-Occupancy, the numbers of sequences for training and evaluation are 23930 and 5119, respectively. For Lyft-Level5, 15720 and 5880 sequences are used for training and evaluation, respectively. The total length of the sequence is set as 7, including 3 frames as observations (2 past frames and 1 present frame) and 4 future frames for forecasting. The range of the occupancy labels is $[-51.2\ m, 51.2\ m]$ for $x$ and $y$ axes, and $[-5\ m, 3\ m]$ for $z$ axis. The voxel resolution is $0.2\ m$, and the grid size is $(512, 512, 40)$ for occupancy labels. The forecasting performances are reported with different time intervals due to the different annotated frequencies, *i.e.*, 2 Hz for nuScenes and nuScenes-Occupancy, and 5 Hz for Lyft-Level5.

**Evaluation Protocol and Metrics.** To fully evaluate the forecasting performance, we follow Cam4DOcc to adopt the following three-level **camera-only** occupancy forecasting tasks: (1) **Forecasting inflated general movable objects (GMO)**: the categories for occupancy grids within bounding boxes from nuScenes and Lyft-Level5 datasets are defined as GMO, while the remaining categories are defined as others. (2) **Forecasting fine-grained GMO**: the same category definition as (1), while the bounding box labels of GMO are replaced with fine-grained voxel-wise annotations from the nuScenes-Occupancy dataset. (3) **Forecasting fine-grained GMO and fine-grained general static objects (GSO)**: the labels of GMO and GSO are from fine-grained annotations. For Lyft-Level5, the evaluation is only conducted on task (1) due to the lack of fine-grained occupancy annotations. For all tasks, intersection over union (IoU (%)) is adopted as the evaluation metric to evaluate occupancy estimation performance for each category of the current frame ($IoU_c$), and any of future frames ($IoU_f$), and the whole span ($\tilde{IoU}_f$). More details can be found in Cam4DOcc.

**Implementation Details.** The proposed OccProphet takes 6 images with $448 \times 800$ pixels from different surround views. Following Cam4DOcc, we use ResNet (He et al., 2016) pre-trained on ImageNet (Deng et al., 2009) with FPN (Lin et al., 2017) as the image encoder. For ablation studies, we use ResNet18 for efficiency, while ResNet34 with deformable convolution (Dai et al., 2017) is adopted for main results. All models are trained with a batch size of 4 on 4 RTX 4090 GPUs and tested on a single RTX 4090 with 24G memory. AdamW (Loshchilov & Hutter, 2019) optimizer with an initial learning rate of 3e-4 and a weight decay of 0.01 is adopted to train the models.

### 4.2 MAIN RESULTS

**Evaluation on forecasting inflated GMO.** The results of forecasting inflated GMO are listed in Table 2, with five comparison methods in total: OpenOccupancy-C, SPC, PowerBEV-3D, BEVDet4D, and OCFNet (Cam4DOcc). OccProphet achieves higher performance than all the compared approaches. Specifically, on the nuScenes dataset, OccProphet surpasses the BEV-based approaches PowerBEV-3D and BEVDet4D by 2.76-11.28%, 2.07-5.69%, 2.28-7.29% in $IoU_c$, $IoU_f$, $\tilde{IoU}_f$ respectively. OccProphet also surpasses the voxel-based method Cam4DOcc in all the metrics, especially in $\tilde{IoU}_f$ by relatively 4.18% and 15.92% on nuScenes and Lyft-Level5 datasets respectively. Qualitative results in Figure 7 demonstrate the superiority of OccProphet. The first group indicates OccProphet's adaptability in low-light conditions.

Table 2: Performance on forecasting inflated GMO. SPC: SurroundDepth (Wei et al., 2023a) + PCPNet (Luo et al., 2023) + Cylinder3D (Zhu et al., 2021).

| Method | nuScenes | | | Lyft-Level5 | | |
|---|---|---|---|---|---|---|
| | $IoU_c$ | $IoU_f$ (2 s) | $\tilde{IoU}_f$ | $IoU_c$ | $IoU_f$ (0.8 s) | $\tilde{IoU}_f$ |
| OpenOccupancy-C (Wang et al., 2023b) | 12.17 | 11.45 | 11.74 | 14.01 | 13.53 | 13.71 |
| SPC (Luo et al., 2023; Wei et al., 2023a; Zhu et al., 2021) | 1.27 | - | - | 1.42 | - | - |
| PowerBEV-3D (Li et al., 2023a) | 23.08 | 21.25 | 21.86 | 26.19 | 24.47 | 25.06 |
| BEVDet4D (Huang & Huang, 2022) | 31.60 | 24.87 | 26.87 | - | - | - |
| OCFNet (Cam4DOcc) (Ma et al., 2024a) | 31.30 | 26.82 | 27.98 | 36.41 | 33.56 | 34.60 |
| OccProphet (ours) | **34.36** | **26.94** | **29.15** | **43.38** | **37.92** | **40.11** |

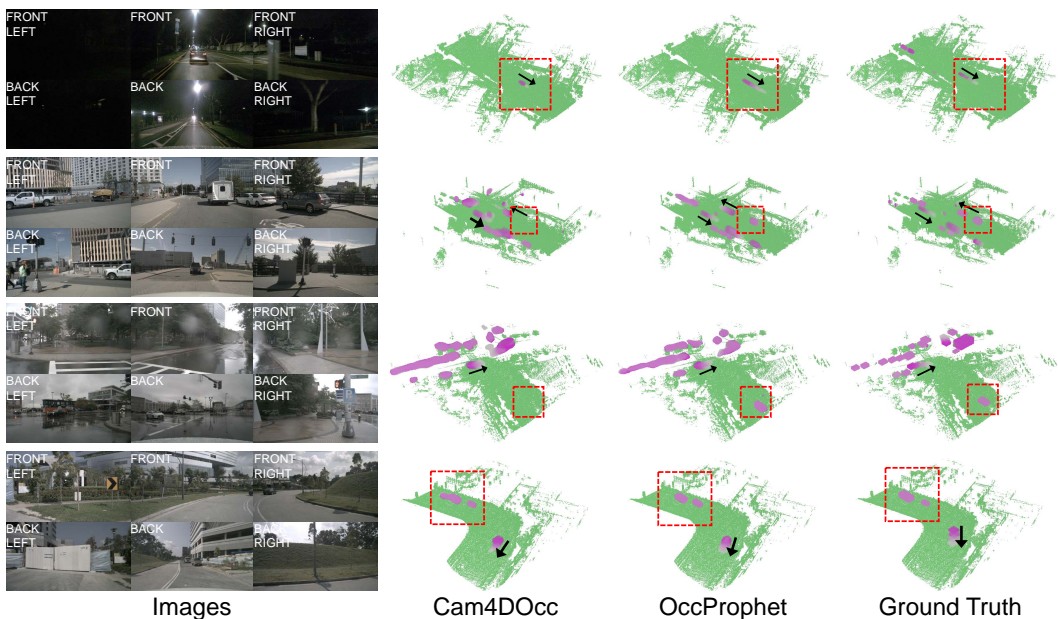

Figure 7: Qualitative results of Cam4DOcc and OccProphet in the future 2 seconds. Black arrows denote the motion trends of moving objects. Red dashed rectangles represent that the results of OccProphet are more consistent with the ground truth than those of Cam4DOcc.

**Evaluation on forecasting fine-grained GMO.** The second task evaluates the performance of forecasting fine-grained GMO. As shown in Table 3, when changing the GMO labels from inflated format using bounding boxes to fine-grained voxel-based labels, the performances of all approaches decrease significantly except the point cloud prediction method SPC achieving slightly better performance compared to Table 2. As pointed out in Cam4DOcc, the reason is that the fine-grained 3D structures of the moving objects are difficult to estimate using past and current continuous RGB images, while the labels for training SPC are also fine-grained and sparse. However, SPC still performs worse than other counterparts due to the lack of shape consistency. Cam4DOcc keeps the lead over other competitors. Despite the challenging task, OccProphet surprisingly precedes all other approaches by a large margin. Specifically, the performances of OccProphet in $IoU_c$, $IoU_f$, $I\tilde{o}U_f$ are relatively 34.3%, 10.43%, 18.61% higher than those of Cam4DOcc. The above results demonstrate the effectiveness of OccProphet in capturing the intricate 3D details of moving objects.

Table 3: Performance on forecasting fine-grained GMO.

| Method | nuScenes-Occupancy | | |
|---|---|---|---|
| | $IoU_c$ | $IoU_f$ (2 s) | $I\tilde{o}U_f$ |
| OpenOccupancy-C (Wang et al., 2023b) | 10.82 | 8.02 | 8.53 |
| SPC (Luo et al., 2023; Wei et al., 2023a; Zhu et al., 2021) | 5.85 | 1.08 | 1.12 |
| PowerBEV-3D (Li et al., 2023a) | 5.91 | 5.25 | 5.49 |
| OCFNet (Cam4DOcc) (Ma et al., 2024a) | 11.45 | 9.68 | 10.10 |
| OccProphet (ours) | **15.38** | **10.69** | **11.98** |

Table 4: Performance on forecasting fine-grained GMO and fine-grained GSO.

| Method | $IoU_c$ | | | $IoU_f$ (2 s) | | | $I\tilde{o}U_f$ |
|---|---|---|---|---|---|---|---|
| | GMO | GSO | mean | GMO | GSO | mean | GMO |
| OpenOccupancy-C (Wang et al., 2023b) | 9.62 | 17.21 | 13.42 | 7.41 | 17.30 | 12.36 | 7.86 |
| SPC (Luo et al., 2023; Wei et al., 2023a; Zhu et al., 2021) | 5.85 | 3.29 | 4.57 | 1.08 | 1.40 | 1.24 | 1.12 |
| PowerBEV-3D (Li et al., 2023a) | 5.91 | - | - | 5.25 | - | - | 5.49 |
| OCFNet (Cam4DOcc) (Ma et al., 2024a) | 11.02 | 17.79 | 14.41 | 9.20 | 17.83 | 13.52 | 9.66 |
| OccProphet (ours) | **13.71** | **24.42** | **19.06** | **9.34** | **24.56** | **16.95** | **10.33** |

**Evaluation on forecasting fine-grained GMO and fine-grained GSO.** The performances of forecasting fine-grained GMO and fine-grained GSO, are listed in Table 4. OccProphet still surpasses Cam4DOcc in all metrics.

### 4.3 ABLATION STUDIES

**Effectiveness of each component.** Table 5 demonstrates the ablation studies on the effectiveness of each component. Removing the Observer (Row 3) leads to around 1% drop in $\tilde{IoU}_f$, showing the importance of the Observer extracting spatio-temporal information from observations. If the Forecaster is removed (Row 4), the performance decreases by 0.74%, indicating the advantage of predicting future states adaptively based on the traffic environment compared to direct prediction. Row 5 shows that removing the Refiner brings a 0.8% drop. If all the components are removed (the last row), where only a single linear layer is used to predict future states, the performances are sharply reduced to 26.07%, further revealing the effectiveness of the components. Qualitative results of using the Observer or not are in Figure 8. The integration of Observer promotes spatio-temporal representativeness, enhancing the forecasting consistency with the ground truth. Other qualitative results are shown in the appendix.

Table 5: Effectiveness of each component.

| Method | $\tilde{IoU}_f$ |
|---|---|
| OccProphet | **28.24** |
| w/o Observer | 27.25 |
| w/o Forecaster | 27.50 |
| w/o Refiner | 27.44 |
| w/o All | 26.07 |

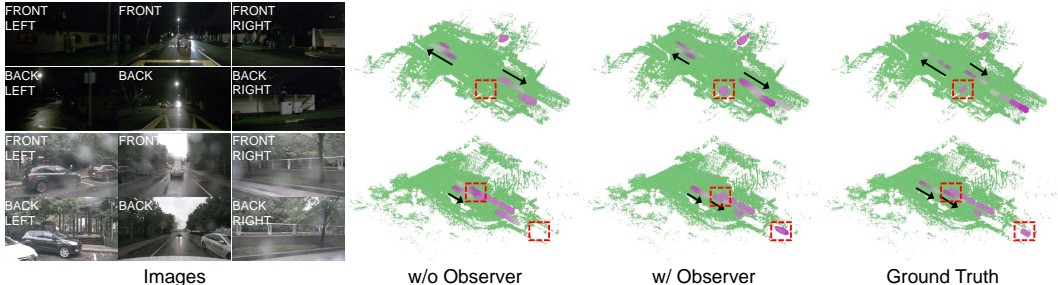

Figure 8: Qualitative results of using the Observer or not. Black arrows denote the motion trends of moving objects. Red dashed rectangles represent that the results with the Observer are more consistent with the ground truth than those without the Observer.

**Effectiveness of E4A representation.** To further justify the effectiveness of E4A representation, we experiment with different representation styles of the Observer. As shown in Table 1, the UNet like E4A balances computational cost and performance well.

### 4.4 COMPARISON OF MODEL COMPLEXITY

In this section, we compare the model complexity between Cam4DOcc and OccProphet. As shown in Table 6, the number of parameters, memory usage, and FLOPs of OccProphet are decreased

Table 6: Comparison of model complexity. $N_p$: Total number of parameters. Mem.: Occupied GPU memory during training.

| Method | $N_p$ (M) | Mem. (G) | FLOPs (G) | FPS | $\tilde{IoU}_f$ |
|---|---|---|---|---|---|
| Cam4DOcc | 370 | 57 | 6434 | 1.7 | 27.98 |
| OccProphet | **82(78%↓)** | **24(58%↓)** | **1985(69%↓)** | **4.5(165%↑)** | **29.15 (4%↑)** |

by 58-78% compared to Cam4DOcc, while OccProphet achieves a 4% relative increase in $\tilde{IoU}_f$, and has 2.6× the FPS speed of Cam4DOcc, justifying the efficiency and effectiveness of OccProphet.

## 5 CONCLUSION

This paper proposes OccProphet, a novel camera-only framework for occupancy forecasting. The framework employs an Observer-Forecaster-Refiner pipeline, specifically designed for efficient and effective training and inference. Such efficiency and effectiveness are achieved through 4D aggregation and tripling-attention fusion on reduced-resolution features. Experimental results demonstrate OccProphet's superiority in both forecasting accuracy and efficiency. It outperforms the state-of-the-art Cam4DOcc in occupancy forecasting by relatively 4%∼18% on three datasets, while operating

2.6× faster and reducing computational costs by 58%-78%. We hope OccProphet can motivate future research in efficient occupancy forecasting and its applications in on-vehicle deployment.

## ACKNOWLEDGEMENTS

The research work was conducted in the JC STEM Lab of Machine Learning and Computer Vision funded by The Hong Kong Jockey Club Charities Trust and was partially supported by the Research Grants Council of the Hong Kong SAR, China (Project No. PolyU 15215824).

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

# A APPENDIX

## A.1 ADDITIONAL QUANTITATIVE RESULTS

**Forecasting inflated GMO and fine-grained GSO**: the category definitions of GMO and GSO are the same as those in the task of **Forecasting fine-grained GMO**. The labels of inflated GMO are generated from bounding boxes, while the labels of fine-grained GSO are occupancy annotations. The performances of forecasting inflated GMO and fine-grained GSO are listed in Table 7. SPC is still the worst, where the IoU of inflated GMO remains consistent in Table 2. OccProphet dramatically outperforms other approaches including OpenOccupancy-C and Cam4DOcc by a large margin. Regarding GMO, OccProphet is 3.77% and 0.92% higher than Cam4DOcc in the current and future moments, respectively. For GSO, OccProphet surpasses Cam4DOcc by 6.46% and 6.38% in $IoU_c$ and $IoU_f$, respectively. When taking multiple future frames for evaluation, OccProphet remains the best with an impressive superiority of 2.21% over the second best Cam4DOcc in $\tilde{IoU}_f$. These results demonstrate the superiority of our method of extracting spatio-temporal features and predicting future states.

Table 7: Performance on forecasting inflated GMO and fine-grained GSO.

| Method | IoU$_c$ | | | IoU$_f$ (2 s) | | | $\tilde{\text{IoU}}_f$ |
|---|---|---|---|---|---|---|---|
| | GMO | GSO | mean | GMO | GSO | mean | GMO |
| OpenOccupancy-C (Wang et al., 2023b) | 13.53 | 16.86 | 15.20 | 12.67 | 17.09 | 14.88 | 12.97 |
| SPC (Luo et al., 2023; Wei et al., 2023a; Zhu et al., 2021) | 1.27 | 3.29 | 2.28 | - | 1.40 | - | - |
| PowerBEV-3D (Li et al., 2023a) | 23.08 | - | - | 21.25 | - | - | 21.86 |
| OCFNet (Cam4DOcc) (Ma et al., 2024a) | 29.84 | 17.72 | 23.78 | 25.53 | 17.81 | 21.67 | 26.53 |
| OccProphet (ours) | **33.61** | **24.18** | **28.89** | **26.45** | **24.19** | **25.32** | **28.74** |

## A.2 QUALITATIVE RESULTS OF ABLATION STUDIES

**Effectiveness of the Forecaster.** Figure 9 shows the qualitative results with or without using the Forecaster module. The Forecaster is adept at perceiving moving objects, while forecasting with only a single linear layer (w/o Forecaster) tends to miss some moving objects.

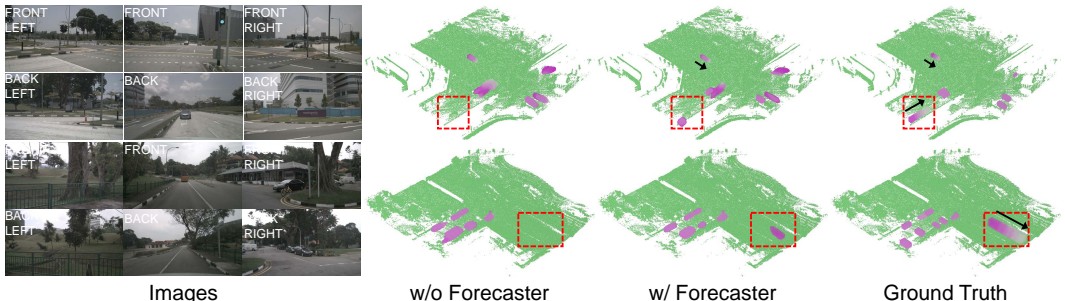

Images      w/o Forecaster      w/ Forecaster      Ground Truth

Figure 9: Qualitative results of using the Forecaster or not. Black arrows denote the motion trends of moving objects. Red dashed rectangles represent that the results with the Forecaster are more consistent with the ground truth than those without the Forecaster.

**Effectiveness of the Refiner.** Figure 10 shows the qualitative results using the Refiner or not. It is evident that using the Refiner yields better forecast results for both moving and static objects, indicating its forecast refinement capability.

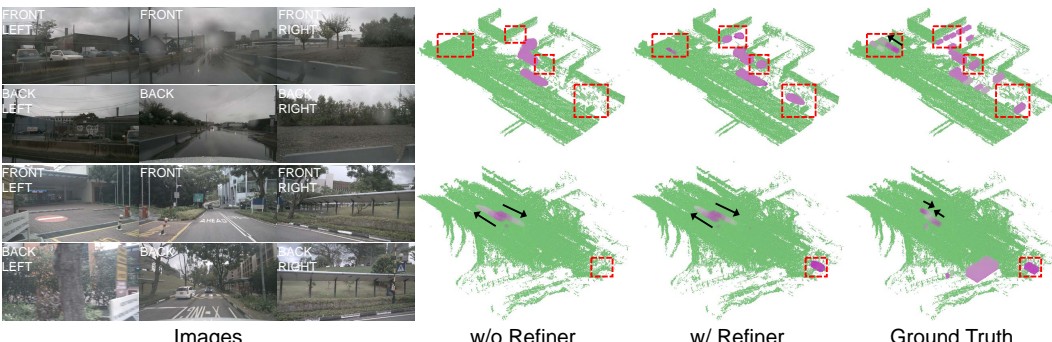

Images      w/o Refiner      w/ Refiner      Ground Truth

Figure 10: Qualitative results of using the Refiner or not. Black arrows denote the motion trends of moving objects. Red dashed rectangles represent that the results with the Refiner are more consistent with the ground truth than those without the Refiner.

## A.3 DISTINCTIONS FROM THE TRADITIONAL ENCODER-DECODER ARCHITECTURE

The traditional encoder-decoder architecture comprises an encoder for representation extraction and a decoder for occupancy prediction, as adopted by OccNet (Tong et al., 2023) and Cam4DOcc. However, the traditional architecture either loses 3D geometry details-*e.g.*, OccNet adopts a BEV-based encoder, or introduces high computational cost-*e.g.*, Cam4DOcc utilizes vanilla 3D convolutional networks as the encoder and decoder.

In OccProphet, the Observer works similarly to an encoder, while the combination of Forecaster and Refiner works similarly to a decoder. Unlike the traditional encoder-decoder architecture, OccProphet pushes the efficiency frontier of 4D occupancy forecasting. To achieve this, the Observer and Refiner enable spatio-temporal interaction using the Efficient 4D Aggregation module, and the Forecaster adaptively predicts future states using a lightweight condition mechanism. Overall, our Observer-Forecaster-Refiner framework emphasizes 4D spatio-temporal interaction and conditional forecasting, meanwhile maintaining efficiency.

## A.4 OCCUPANCY FORECASTING OVER VARYING TIME HORIZONS

The performance of occupancy forecasting over varying time horizons is critical in autonomous driving scenarios. We compare OccProphet with OpenOccupancy-C (Wang et al., 2023b), PowerBEV-3D (Li et al., 2023a), and Cam4DOcc (Ma et al., 2024a) in terms of occupancy forecasting accuracy over varying time horizons, as shown in Table 8. We can see that (1) OccProphet consistently outperforms other approaches across all time horizons on three datasets. (2) The longer the forecasting period, the lower the accuracy of all methods, indicating that forecasting becomes increasingly difficult.

Table 8: Performance on occupancy forecasting over varying time horizons.

| Method | nuScenes | | | | Lyft-Level5 | | | | nuScenes-Occupancy | | | |
|---|---|---|---|---|---|---|---|---|---|---|---|---|
| | 0.5s | 1.0s | 1.5s | 2.0s | 0.2s | 0.4s | 0.6s | 0.8s | 0.5s | 1.0s | 1.5s | 2.0s |
| OpenOccupancy-C | 12.07 | 11.80 | 11.63 | 11.45 | 13.87 | 13.77 | 13.65 | 13.53 | 9.17 | 8.64 | 8.29 | 8.02 |
| PowerBEV-3D | 22.48 | 22.07 | 21.65 | 21.25 | 25.70 | 25.25 | 24.82 | 24.47 | 5.74 | 5.56 | 5.41 | 5.25 |
| OCFNet (Cam4DOcc) | 29.36 | 28.30 | 27.44 | 26.82 | 35.58 | 34.96 | 34.28 | 33.56 | 10.64 | 10.20 | 9.89 | 9.68 |
| **OccProphet (ours)** | **32.17** | **29.60** | **27.95** | **26.94** | **42.34** | **40.87** | **39.38** | **37.92** | **13.64** | **12.10** | **11.27** | **10.69** |

## A.5 FAILURE CASES

### A.5.1 UNSEEN SCENARIOS

To test forecasting performance in unseen scenarios, we conduct a cross-domain experiment. Specifically, we train the model on the Lyft-Level5 dataset and test it on the nuScenes dataset. The visualization results are shown in Figure 11. We can observe that cross-domain occupancy forecasting performs worse than intra-domain forecasting within a single dataset. We consider that this performance gap may be due to the difference in sensor settings between the two datasets. In the future, research on generalized occupancy forecasting will be a valuable direction worth exploring.

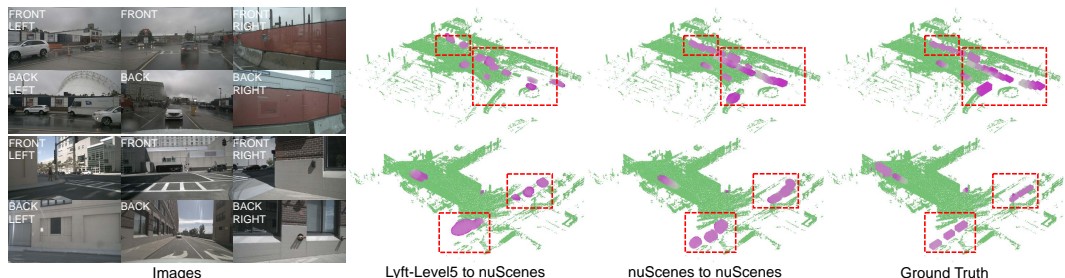

Figure 11: Failure cases in unseen scenarios. Red dashed rectangles indicate that the intra-domain results (*e.g.*, nuScenes to nuScenes) are more consistent with the ground truth than cross-domain occupancy forecasting (*e.g.*, Lyft-Level5 to nuScenes).

### A.5.2 OCCLUDED SCENARIOS

We investigate the forecasting performance in occluded scenarios, which frequently occur in traffic scenarios. Ground truth occupancy and forecasted future occupancy by OccProphet, over varying time horizons, are qualitatively visualized in Figure 12. It can be observed that the occupancy labels of the occluded object over different time horizons are complete, ensuring the training reliability. The forecasted results of OccProphet demonstrate its capability of handling occluded scenarios to a certain degree, which can still be further improved in future research.

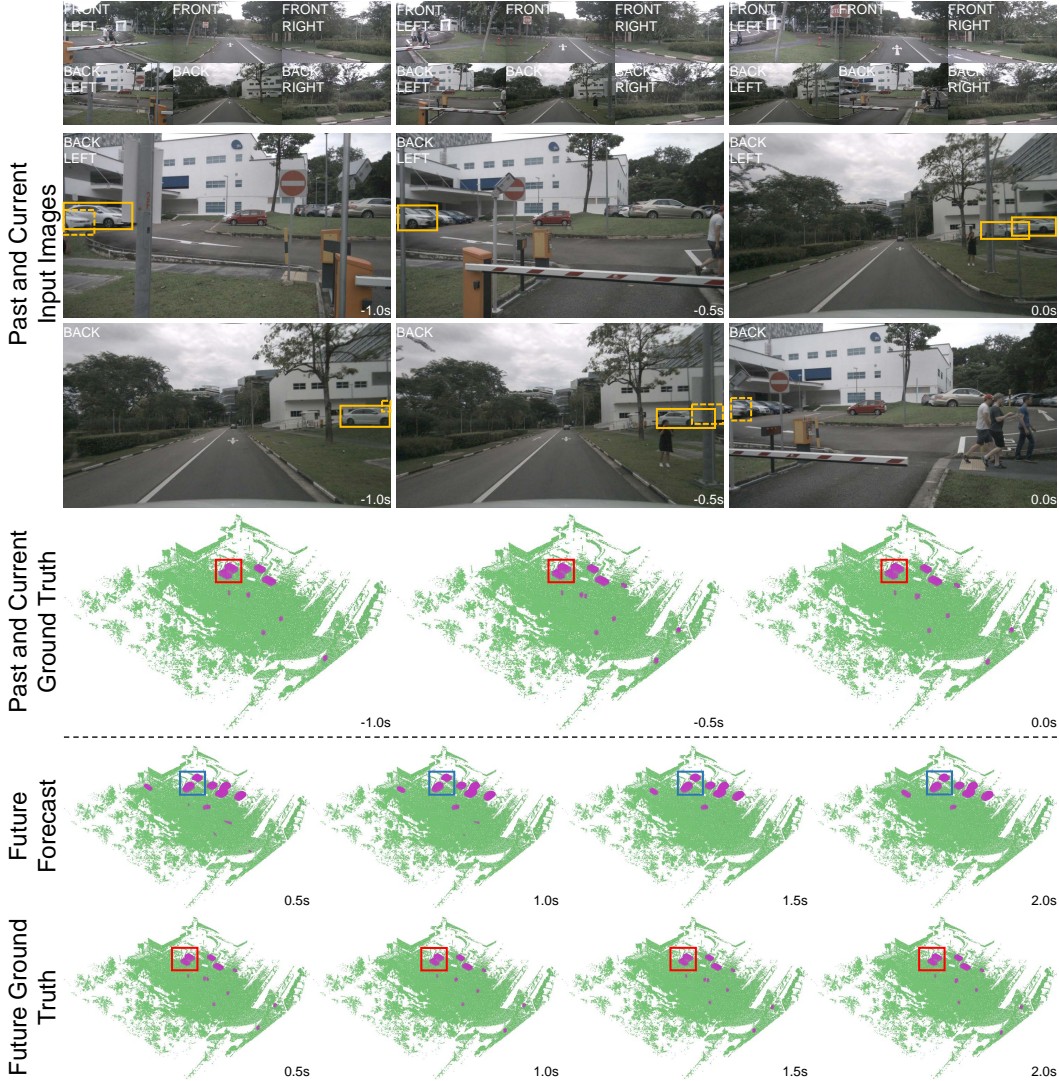

Figure 12: Failure cases in occluded scenarios. In input images, yellow rectangles indicate the object of interest, while dashed rectangles denote that the object is nearly entirely occluded. Red rectangles indicate that the occupancy ground truths of the occluded object are complete over past, current, and future frames. Blue rectangles reveal that OccProphet is able to forecast the future occupancy of the occluded object, which can mitigate the occlusion issue to a certain extent.

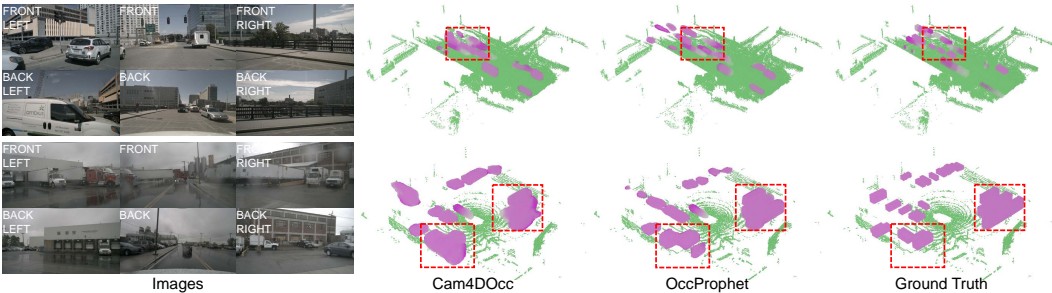

Figure 13: Failure cases in dense scenarios. Red dashed rectangles indicate that the results of OccProphet are more consistent with the ground truth than those of Cam4DOcc.

### A.5.3 DENSE SCENARIOS

To evaluate model precision at fine granularity, we qualitatively visualize the occupancy forecasting of Cam4DOcc and OccProphet in dense scenarios, as shown in Figure 13. The visualization reveals that both Cam4DOcc and OccProphet encounter challenges in fine-grained forecasting, indicating a huge difficulty. In comparison, OccProphet's results are closer to the ground truth. We believe that this is attributed to our proposed Observer-Forecaster-Refiner framework. We regard fine-grained occupancy forecasting as a valuable direction for future research.

