# OpenReview forum: "OccProphet: Pushing the Efficiency Frontier of Camera-Only 4D Occupancy Forecasting with an Observer-Forecaster-Refiner Framework"
_ICLR.cc/2025/Conference — ICLR 2025 Poster_

### Official Review · Reviewer_t8d8 · 2024-10-30

**Soundness:** 3
**Presentation:** 3
**Contribution:** 3
**Rating:** 6
**Confidence:** 3

**Summary:**

The paper introduces a novel framework named OccProphet, designed to predict the 3D occupied status of driving environments based on historical 2D images, with a focus on improving efficiency and reducing computational demands during training and inference stages. This is particularly aimed at enhancing the feasibility of deploying such technology on edge agents like autonomous vehicles.The OccProphet framework consists of three lightweight components: the Observer, Forecaster, and Refiner. The Observer extracts spatio-temporal features from 3D data using an Efficient 4D Aggregation with Tripling-Attention Fusion method. The Forecaster and Refiner work together to conditionally predict and refine future occupancy inferences. The paper claims that OccProphet is both training- and inference-friendly, reducing computational costs by 58% to 78% with a 2.6x speedup compared to the state-of-the-art Cam4DOcc method, while achieving 4% to 18% higher forecasting accuracy. The experimental results are demonstrated using the nuScenes, Lyft-Level5, and nuScenes-Occupancy datasets.

**Strengths:**

1. The paper introduces a new framework, OccProphet, which is designed to be efficient and effective for camera-only 4D occupancy forecasting, a critical capability for autonomous driving.
2. The framework significantly lowers computational requirements by 58% to 78% compared to the state-of-the-art Cam4DOcc, making it more feasible for deployment on edge agents like autonomous vehicles.
3. OccProphet achieves a relative increase in forecasting accuracy of 4% to 18% over existing methods, which is a substantial improvement in the field of autonomous driving perception.

**Weaknesses:**

The paper does not give detailed explanations for the design of the three proposed modules.

**Questions:**

What is the actual frames-per-second of the model. And is it good enough for deployment?

---

> ### Author Response · Authors · 2024-11-25
> **Official Comment by Authors**
>
> Thanks for your positive recognition (_e.g._, **critical for autonomous driving**, **more feasible for deployment**, and **substantial improvement in perception**). We appreciate the opportunity to address your valuable feedback as follows.
>
> ## **Response to Weaknesses**
> ***
> **W1. The paper does not give detailed explanations for the design of the three proposed modules.**
>
> Thanks for pointing this out. We further provide more explanations for the design of the three proposed modules in the following perspectives. The manuscript has been updated accordingly.
>
> - Overview:
>
>   "The Observer module efficiently and effectively aggregates spatio-temporal information within multi-frame observations (*i.e.*, multi-frame 3D voxel features). The Observer’s output then undergoes a Forecaster, which adaptively predicts future states, ensuring flexibility across diverse traffic conditions. The Refiner module further enhances the quality of these predictions by enabling cross-frame interactions." (In Para. 1 of Sec. 3.1)
>
>
> - Observer:
>
>   Efficient 4D Aggregation (E4A): "Directly aggregating the original 4D feature $F_{motion}$ will incur a high computational cost. For efficiency, we design the Efficient 4D Aggregation (E4A) module to first produce compact features through downsampling, and then exploit spatio-temporal interactions on the compact features to achieve aggregation, followed by the upsampling process to compensate for the information loss." (In Para. 1 of Sec. 3.2.1)
>
>   Tripling-Attention Fusion (TAF): "The tripling operation is designed to understand the 3D space from three complementary and compact perspectives, which can retain 3D scene information with fewer computational costs. Specifically, a tripling operation decomposes a 3D feature into three distinct branches: scene, height, and BEV." (In Para. 2 of Sec. 3.2.2)
>
> - Forecaster:
>
>   "We propose forecasting occupancy with flexibility to adapt to various traffic scenarios featuring diverse spatio-temporal complexities. To achieve this, we design a novel Forecaster module that predicts future states based on the overall environmental condition. The Forecaster comprises a Condition Generator and a Conditional Forecaster." (In Para. 2 of Sec. 3.3)
>
> - Refiner:
>
>   "Since the Forecaster module predicts $F_{\text{future}}$ using linear projection, it inevitably lacks cross-frame interactions. The Refiner is designed to enhance the forecasted results via further interactions between future frames, as well as incorporating historical frames as supplementary information." (In Para. 1 of Sec. 3.4)
>
> Besides, we added some technical details, highlighted in blue in Sec. 3 of the updated paper.
>
> ## **Response to Questions**
> ***
>
> **Q1. What is the actual frames-per-second of the model? And is it good enough for deployment?**
>
> After engineering optimization, the inference speed of Cam4DOcc is accelerated from 1.7 FPS to 8.0 FPS. In contrast, OccProphet achieves a speedup from 4.5 FPS to 21.2 FPS, which is close to deployment requirements.

---

> > ### Comment · Reviewer_t8d8 · 2024-11-27
> >
> > The reviewer appreciates the clarification from the authors and prefer maintaining the score.

---

> > > ### Author Response · Authors · 2024-11-27
> > > **Official Comment by Authors**
> > >
> > > Thank you sincerely for your professional review and valuable comments. We appreciate your positive feedback and decision to maintain the score. If you have any further questions, feel free to discuss with us.

---

### Official Review · Reviewer_ZdEq · 2024-11-01

**Soundness:** 3
**Presentation:** 3
**Contribution:** 3
**Rating:** 6
**Confidence:** 4

**Summary:**

This paper propose a method called OCCPROPHET for efficient occupancy forecasting with camera-only inputs. The proposed framework is consisted of Observer, Forecaster, and Refiner. OCCPROPHET first embeds the sequence of camera images into 3D voxel features. Then Observer applies 4D feature aggregation to gradually reduce the spatial resolution of the 3D features and a tripling-attention fusion strategy on the lower-resolution 3D features to reduce information loss. The Forecaster component forecast the state of the environment based on the feature output from Observer. Finally, Refiner utilize temporal relationship to enhance the quality of the 3D voxel features to generate the final prediction of future occupancy. The key idea is reducing spatial resolution at the very beginning of the network to reduce computational cost during embedding and forecasting. The tripling-attention part take spatiotemporal interactions into account to reduce the information loss and the Refiner part makes use of temporal correspondence to increase the granularity of the prediction. The proposed method makes a good trade-off between efficiency and effectiveness. The extensive experiment results show that OCCPROPHET largely reduce computational cost while achieving slightly better performance.

**Strengths:**

This paper propose OCCPROPHET to improve the efficiency of occupancy forecasting algorithm while maintaining and even improving the performance. The key idea is to embed a coarse-level 3D voxel features at the very beginning of the network to reduce computational cost and meanwhile utilize the temporal-spatial relationship to reduce information loss during embedding and forecasting. Finally, in order to guarantee prediction quality, the Refiner part enhance the feature quality with temporal-spatial correspondence and increase the granularity of the prediction. OCCPROPHET provides a better balance between efficiency and effectiveness in occupancy forecasting problem. The experiments are extensive and the results demonstrate both the efficiency and effectiveness of OCCPROPHET.

**Weaknesses:**

The reviewer is concerned about the effectiveness part of OCCPROPHET. Since OCCPROPHET reduce spatial resolution at the beginning of the network, there should be information loss no matter what strategy OCCPROPHET uses to compensate that. At the same time, temporal-spatial interaction should also be utilized in OCFNet (Cam4DOcc). The reviewer is a bit confused about the better performance OCCPROPHET shows than that of OCFNet (Cam4DOcc). The reviewer is concerned about whether the performance improvement stems from the fact that OCFNet (Cam4DOcc) is underfit and OCCPROPHET converges faster.

**Questions:**

1) Could the authors show experiment results on training for longer with both OCFNet (Cam4DOcc) and OCCPROPHET to see whether the performance improvement stems from early convergence.

2) If not, is there any analysis or explanation about the performance improvement with respect to the information loss?

---

> ### Author Response · Authors · 2024-11-25
> **Official Comment by Authors**
>
> Thanks for your in-depth review, endorsement of our work (_e.g._, **extensive experiments** and **an efficient and effective algorithm**), and constructive advice. Below, we provide detailed responses to each of your points.
>
> ## **Response to Weaknesses**
> ***
> **W1. The reviewer is concerned about the effectiveness part of OCCPROPHET. The temporal-spatial interaction should also be utilized in OCFNet (Cam4DOcc). The reviewer is a bit confused about the better performance OCCPROPHET shows than that of OCFNet (Cam4DOcc).**
>
> | Method | $\text{IoU}_{c}$ | $\tilde{\text{IoU}}_{f}$ |
> |:---:|:---:|:---:|
> | Cam4Docc | 31.30 |  27.98 |
> | Cam4DOcc w/ temporal-spatial interaction | 33.63 |  28.29 |
> | **OccProphet (ours)** | **34.36** |  **29.15** |
>
> The occupancy forecasting performance of OCFNet (Cam4DOcc), with and without temporal-spatial interaction, is presented in the table above. The integration of temporal-spatial interaction improves Cam4DOcc, yielding higher scores in $\text{IoU} _ {c}$ (accuracy of the current frame) and $\tilde{\text{IoU}}_{f}$ (average accuracy of all future frames), demonstrating the effectiveness of this interaction. In comparison, OccProphet outperforms Cam4DOcc, even with temporal-spatial interaction.
>
> We further conduct an experiment where both OCFNet (Cam4DOcc) and OccProphet are trained for longer until convergence. The experimental results and analysis are provided in the response to Question 1 below.
>
> ## **Response to Questions**
> ***
>
> **Q1. Could the authors show experiment results on training for longer with both OCFNet (Cam4DOcc) and OCCPROPHET to see whether the performance improvement stems from early convergence.**
>
> We train OCFNet (Cam4DOcc) and OccProphet for longer until they both converge (36 epochs).
>
> The quantitative comparison of occupancy forecasting performance is presented in the table below. We observe that both OCFNet (Cam4DOcc) and OccProphet benefit from longer training. However, OccProphet still outperforms Cam4DOcc, indicating that the performance improvement does not stem from early convergence.
>
> | Method | Epochs | $\text{IoU}_{c}$ | $\text{IoU}_{f}\ (0.5s)$ | $\text{IoU}_{f}\ (1.0s)$ | $\text{IoU}_{f}\ (1.5s)$ | $\text{IoU}_{f}\ (2.0s)$ | $\tilde{\text{IoU}}_{f}$ |
> |:---:|:---:|:---:|:---:|:---:|:---:|:---:|:---:|
> | Cam4DOcc | 24 (original) | 31.30 | 29.36 | 28.30 | 27.44 | 26.82 | 27.98 |
> | **OccProphet** | 24 (original) | **34.36** | **32.17** | **29.60** | **27.95** | **26.94** | **29.15** |
> | Cam4DOcc | 36 | 33.86 | 31.71 | 29.09 | 27.39 | 26.40 | 28.65 |
> | **OccProphet** | 36 | **34.85** | **32.46** | **29.74** | **27.96** | **26.70** | **29.19** |
>
> **Q2. If not, is there any analysis or explanation about the performance improvement?**
>
> The performance improvement stems from the collaborative functions of the three modules within our proposed Observer-Forecaster-Refiner framework.
>
> - Temporal-spatial Interaction Mechanism: This interaction is embedded in the Observer and Refiner modules. As analyzed in Weakness 1, this mechanism has been proven to enhance forecasting accuracy.
>
> - Condition Mechanism: The Forecaster module predicts future states based on the overall environmental condition. This mechanism enables forecasting to adapt to various traffic scenarios.
>
> - Independent Module Contributions: As shown in Tab. 5 and Fig. 8, 9, and 10 of our paper, each module—Observer, Forecaster, and Refiner—is important to improve occupancy forecasting accuracy.
>
> In summary, the performance improvement is attributed to our proposed Observer-Forecaster-Refiner framework. This framework is not only lightweight but also effective in enhancing performance.

---

> > ### Comment · Reviewer_ZdEq · 2024-11-26
> > **Thank you for the response.**
> >
> > The reviewer appreciate the response from the authors and want to maintain the score.

---

> > > ### Author Response · Authors · 2024-11-27
> > > **Official Comment by Authors**
> > >
> > > Thank you for your patience and expertise during the review. We appreciate your decision to maintain the positive assessment. If you have any further questions, feel free to discuss with us.

---

### Official Review · Reviewer_uB9U · 2024-11-01

**Soundness:** 3
**Presentation:** 3
**Contribution:** 3
**Rating:** 8
**Confidence:** 4

**Summary:**

OccProphet is an efficient occupancy forecasting framework for autonomous driving, reducing computational costs by 58%–78% and achieving a 2.6× speedup over existing methods while maintaining high accuracy. Through its Observer, Forecaster, and Refiner modules, OccProphet predicts future 3D occupancy from 2D images, making it suitable for real-time edge deployment.

**Strengths:**

a). OccProphet addresses the computational limitations of previous methods, a crucial improvement for deploying autonomous vehicles on edge devices.

b). The authors’ writing style is clear and concise, effectively conveying the ideas and concepts of 4D occupancy prediction.

**Weaknesses:**

a) Although the authors have distinctively named the three components as Observer, Forecaster, and Refiner to differentiate them from previous methods, the paper should more clearly highlight the distinctions from the traditional encoder-decoder architecture to better emphasize its contribution.

b) The tripling-attention and reduced-resolution feature aggregation may sacrifice some granularity or detail in the forecasts, possibly affecting the precision of the model in dense scenarios.

c) The paper lacks details on how well the model performs over varying forecasting time horizons, especially under extended timeframes, which are critical in autonomous driving scenarios.

**Questions:**

The current baselines lack sufficient comparison methods and evaluation metrics, which are inadequate to demonstrate the superiority of the proposed approach.

---

> ### Author Response · Authors · 2024-11-25
> **Official Comment by Authors (Part 1)**
>
> Thank you for your kind approval (_e.g._, **a crucial improvement for deploying** and **good writing**). Below, we provide detailed responses to your insightful comments.
>
> ## **Response to Weaknesses**
> ***
>
> **W1. The paper should more clearly highlight the distinctions from the traditional encoder-decoder architecture to better emphasize the contribution of Observer, Forecaster, and Refiner.**
>
> Thanks for the constructive comment. Based on the suggestion, we have added the distinction statement in the updated paper. "The traditional encoder-decoder architecture comprises an encoder for representation extraction and a decoder for occupancy prediction, as adopted by OccNet [1] and Cam4DOcc. However, the traditional architecture either loses 3D geometry details—*e.g.*, OccNet adopts a BEV-based encoder, or introduces high computational cost—*e.g.*, Cam4DOcc utilizes vanilla 3D convolutional networks as the encoder and decoder.
>
> In OccProphet, the Observer works similarly to an encoder, while the combination of Forecaster and Refiner works similarly to a decoder. Unlike the traditional encoder-decoder architecture, OccProphet pushes the efficiency frontier of 4D occupancy forecasting. To achieve this, the Observer and Refiner enable spatio-temporal interaction using the Efficient 4D Aggregation module, and the Forecaster adaptively predicts future states using a lightweight condition mechanism. Overall, our Observer-Forecaster-Refiner framework emphasizes 4D spatio-temporal interaction and conditional forecasting, meanwhile maintaining efficiency." (In Appendix A.3)
>
> [1] Wenwen Tong, Chonghao Sima, Tai Wang, Li Chen, Silei Wu, Hanming Deng, Yi Gu et al. "Scene as occupancy." In Proceedings of the IEEE/CVF International Conference on Computer Vision, pp. 8406-8415. 2023.
>
> **W2. The tripling-attention and reduced-resolution feature aggregation may sacrifice some granularity or detail in the forecasts, possibly affecting the precision of the model in dense scenarios.**
>
> This issue does exist. The visualization of occupancy forecasting in dense scenarios is provided in Fig. 13, with the corresponding analysis in Appendix A.5.3 of the updated paper. It is obvious that both Cam4DOcc and OccProphet have errors in fine-grained forecasting. In comparison, OccProphet's results are closer to the ground truth, while our goal is to achieve high efficiency and reduce performance drop.
>
> We consider that, although our aggregation operates on reduced-resolution features, it is multi-scale and accompanied by feature upsampling. Besides, the tripling-attention splits input features from the scene, height, and BEV perspectives. Among them, the height perspective captures vertical details of the environment, while the BEV perspective effectively obtains a large receptive field through the attention mechanism. Together, these aspects enhance the ability of the tripling-attention to capture scene details.
>
> In Appendix A.5.3, we have claimed that fine-grained occupancy forecasting is a valuable direction for future research.
>
> **W3. The paper lacks details on how well the model performs over varying forecasting time horizons.**
>
> |  | nuScenes |  |  |  | Lyft-Level5 |  |  |  | nuScenes-Occupancy |  |  |  |
> |:---:|:---:|:---:|:---:|:---:|:---:|:---:|:---:|:---:|:---:|:---:|:---:|:---:|
> | Method | 0.5s | 1.0s | 1.5s | 2.0s | 0.2s | 0.4s | 0.6s | 0.8s | 0.5s | 1.0s | 1.5s | 2.0s |
> | OpenOccupancy-C | 12.07 | 11.80 | 11.63 | 11.45 | 13.87 | 13.77 | 13.65 | 13.53 | 9.17 | 8.64 | 8.29 | 8.02 |
> | PowerBEV-3D | 22.48 | 22.07 | 21.65 | 21.25 | 25.70 | 25.25 | 24.82 | 24.47 | 5.74 | 5.56 | 5.41 | 5.25 |
> | OCFNet (Cam4DOcc) | 29.36 | 28.30 | 27.44 | 26.82 | 35.58 | 34.96 | 34.28 | 33.56 | 10.64 | 10.20 | 9.89 | 9.68 |
> | **OccProphet (ours)** | **32.17** | **29.60** | **27.95** | **26.94** | **42.34** | **40.87** | **39.38** | **37.92** | **13.64** | **12.10** | **11.27** | **10.69** |
>
> Thank you for the professional advice. We evaluated the performance of forecasting occupancy over varying time horizons, as shown in the above table. We can see that (1) OccProphet consistently outperforms other approaches across all time horizons on three datasets. (2) The longer the forecasting period, the lower the accuracy of all methods, indicating that forecasting becomes increasingly difficult.
>
> We have included this experiment in Appendix A.4 of the updated paper.

---

> ### Author Response · Authors · 2024-11-25
> **Official Comment by Authors (Part 2)**
>
> ## **Response to Questions**
> ***
>
> **Q1. The current baselines lack sufficient comparison methods and evaluation metrics.**
>
> The previous benchmark in Cam4DOcc covers four camera-only occupancy forecasting tasks (see Sec. 4.2 and Appendix A.1), three datasets (nuScenes, Lyft-Level5, nuScenes-Occupancy), three accuracy metrics ($\text{IoU} _ {c}$, $\text{IoU} _ {f}$, and $\tilde{\text{IoU}} _ {f}$). In our original paper, we have added four efficiency metrics (N$_{\text{p}}$ (M), Memory (G), FLOPs (G), and FPS) for comparisons.
>
> In the updated paper, we have added a quantitative comparison of forecasting over varying time horizons (see Tab. 8) and a qualitative comparison on dense scenarios (see Fig. 13). For the task of occupancy forecasting over varying time horizons, we use $\text{IoU}_{f}$ at different timestamps as the evaluation metric. For example, on the nuScenes and nuScenes-Occupancy datasets, we evaluate forecasting performance at 0.5, 1.0, 1.5, and 2.0 seconds; on the Lyft-Level5 dataset, we evaluate forecasting at 0.2, 0.4, 0.6, and 0.8 seconds.
>
> | Method | $\text{IoU}_{c}$ | $\text{IoU}_{f}\ (2.0s)$ | $\tilde{\text{IoU}}_{f}$ |
> |:---:|:---:|:---:|:---:|
> | OpenOccupancy-C | 12.17 | 11.45 | 11.74 |
> | SPC | 1.27 | - | - |
> | PowerBEV-3D | 23.08 | 21.25 | 21.86 |
> | BEVDet4D [1] | 31.60 | 24.87 | 26.87 |
> | OCFNet (Cam4DOcc) | 31.30 | 26.82 | 27.98 |
> | **OccProphet (ours)** | **34.36** | **26.94** | **29.15** |
>
> Furthermore, we compared a new baseline method, BEVDet4D, on the nuScenes dataset (see the table above). There are five comparison methods totally.
>
> In summary, by expanding the baseline methods, comparison tasks, and evaluation metrics, we provide a more comprehensive demonstration of the superiority of our proposed approach.
>
> [1] Junjie Huang, Guan Huang. "Bevdet4d: Exploit temporal cues in multi-camera 3d object detection." arXiv preprint arXiv:2203.17054, 2022.

---

> > ### Comment · Reviewer_uB9U · 2024-11-26
> >
> > Thank you for the explanation! The reviewer believes that the updated results and explanation can be further added to the revision. Most of the concerns are addressed and the reviewer wants to raise the score to 8.

---

> > > ### Author Response · Authors · 2024-11-27
> > > **Official Comment by Authors**
> > >
> > > Thank you sincerely for your positive recognition of our work. Based on the valuable advice, we have further added updated results and explanations in Tab. 2 and Para. 1 of Sec. 4.2. We are truly grateful for your time and effort in the review!

---

### Official Review · Reviewer_gfQ3 · 2024-11-02

**Soundness:** 4
**Presentation:** 3
**Contribution:** 4
**Rating:** 8
**Confidence:** 5

**Summary:**

This paper introduces OccProphet, a novel camera-only framework for occupancy forecasting. It features an Observer-Forecaster-Refiner pipeline optimized for efficient training and inference, utilizing 4D aggregation and tripling-attention fusion on reduced-resolution features. Experimental results show that OccProphet outperforms existing methods in both accuracy and efficiency.

**Strengths:**

1. The article is well-written and easy-to-follow.
2. Figure 2 is helpful to understand the effect of OccProphet.
3. The ablation study is helpful in demonstrating the benefits of each module in OccProphet.
4. Extensive experimental results on several benchmarks back up the effectiveness of OccProphet.

**Weaknesses:**

1. Some symbols and design details of the model are unclear.
2. The analysis of failure scenarios is lacking, but this is not a major concern.
3. See Questions.

**Questions:**

1. What part of the Cam4DOcc model contributes to its high computational cost? How much memory does Cam4DOcc use during training and testing? Approximately how many GPU days are needed for full training?

2. In multi-frame 3D occupancy results, how can environmental dynamics be effectively captured? The proposed Observer-Forecaster-Refiner pipeline seems to learn the dynamics of objects in the latent space without strict physical constraints. If the real world presents scenarios that are rare or unseen in the dataset, could 4D predictions fail? Are there any examples of such failures?

3. 4D prediction is challenging, especially when occlusions are frequent, leading to potential voxel loss. Have the labels in the 4D dataset been completed to account for these occlusions? If they have, can the authors' method handle situations where any frame from the historical RGB images loses an object?


4. Are the 6-DoF camera parameters in the Observer used to align historical 3D features with the current frame? In lines 188-191, why does F change to F_{motion}, resulting in an extra matrix dimension of 6×X×Y×Z? Is this converting the 6-DoF pose into a matrix?

5. In lines 206-207, how does C+6 become C? Is this done through a 1×1×1 3D convolution?

6. What is the difference between the E4A module shown in Figure 4 and the UNet structure? It looks like a 4D version of UNet. Why does the FLOPs increase significantly while the number of parameters in E4A decreases? Is this due to the upsampling process?

7. In TAF, after applying 3D, 2D, and 1D global average pooling, the same scale features perform temporal attention. Would cross-attention between different scales yield better results?

8. In the Forecaster, is the condition for prediction learned from past voxels?

9. What does the colon symbol in line 313 mean?

10. The comparisons could be more thorough. For example, it would be helpful to follow the output of the 4D BEV method, like [1], with a 2D to 3D detection head for comparison.
[1] Bevdet4d: Exploit temporal cues in multi-camera 3D object detection

---

> ### Author Response · Authors · 2024-11-25
> **Official Comment by Authors (Part 1)**
>
> Thank you for your encouraging appreciation (_e.g._, **well-written manuscript**, **clear Fig. 2**, and **thorough experiments**). Below, we address each point in detail according to your insightful comments and constructive advice.
>
> ## **Response to Weaknesses**
> ***
>
> **W1. Some symbols and design details of the model are unclear.**
>
> For the issues regarding unclear symbols (Question 4, 5, and 9) and unclear design (Question 6 and 8), we have provided detailed explanations. Additionally, we add more explanations for the model design in the paper. All revisions to the model design and technical details are highlighted in blue in Sec. 3 of the updated paper.
>
> **W2. The analysis of failure scenarios is lacking, but this is not a major concern.**
>
> Thanks for your advice. We have added some failure scenarios (shown in Fig. 11, 12, and 13) along with corresponding discussions in Appendix A.5.
>
> ## **Response to Questions**
> ***
>
> **Q1-(1). What part of the Cam4DOcc model contributes to its high computational cost?**
>
>
> The computational cost of each component and the scores of IoU are listed below.
> ($^{\\#}$: Tested by MMDetection3D \[1\]. $^{*}$: Reported from the original paper.)
> | Method | FLOPs of Image Encoder and 2D-to-3D Lifting Module | FLOPs of Voxel Encoder | FLOPs of Prediction Module | FLOPs of Voxel Decoder | FLOPs of Occupancy and Flow Heads | FLOPs of Whole Model | $\tilde{\text{IoU}}_{f}$ |
> |---|:---:|:---:|:---:|:---:|:---:|:---:|:---:|
> | Cam4DOcc  | 4836 | 620 | 665 | 391 | 32 | 6544$^{\\#}$ (6434$^*$) | 27.98 |
>
>
> | Method | FLOPs of Image Encoder and 2D-to-3D Lifting Module | FLOPs of Observer | FLOPs of Forecaster | FLOPs of Refiner | FLOPs of Occupancy and Flow Heads | FLOPs of Whole Model | $\tilde{\text{IoU}}_{f}$ |
> |---|:---:|:---:|:---:|:---:|:---:|:---:|:---:|
> | **OccProphet (ours)**  | 1157 | 271 | <0.1 | 529 | 28 | **1985$^{\\#}$** | **29.15** |
>
> The high computational cost of Cam4DOcc primarily stems from its image encoder and 2D-to-3D lifting module. To alleviate this computational burden, OccProphet reduces the input image size by half. However, this operation will decrease occupancy forecasting accuracy.
>
> To compensate for the loss in forecasting accuracy, meanwhile maintaining efficiency, OccProphet introduces lightweight components: Observers, Forecasters, and Refiners. These components are both effective and efficient:
> (i) The FLOPs of the remaining modules in Cam4DOcc is 1708G. In contrast, with the same input feature size, the remaining part of OccProphet is 828G FLOPs.
> (ii) OccProphet achieves an $\tilde{\text{IoU}}_{f}$ of 29.15%, surpassing Cam4DOcc's 27.98%, thereby indicating higher occupancy forecasting accuracy.
>
> \[1\]: MMDetection3D Contributors. Mmdetection3d: Openmmlab next-generation platform for general 3d object detection. https://github.com/open-mmlab/mmdetection3d, 2020. 1, 4
>
> **Q1-(2). How much memory does Cam4DOcc use during training and testing?**
>
> Cam4DOcc uses 57G of memory during training (as reported in the original paper), and approximately 24G of memory during testing. In comparison, OccProphet requires 24G of memory during training, and around 8G during testing.
>
> **Q1-(3). Approximately how many GPU days are needed for full training?**
>
> The full training time of Cam4Docc is approximately 56 GPU Days on NVIDIA A100. In comparison, OccProphet takes around 19 GPU Days for training on NVIDIA A100.
>
> **Q2. The proposed Observer-Forecaster-Refiner pipeline seems to learn the dynamics of objects in the latent space without strict physical constraints. If the real world presents scenarios that are rare or unseen in the dataset, could 4D predictions fail? Are there any examples of such failures?**
>
> Thank you for the comment. We use occupancy flow labels between adjacent frames as physical constraints in training. These labels ensure that dynamic objects adhere to real-world motion principles, *i.e.*, moving smoothly and continuously in 3D space and time, and avoiding unrealistic changes in velocity or direction.
>
> In the updated paper, we have added several failure cases, including cross-domain forecasting for unseen scenarios, occupancy forecasting for occluded scenarios, and fine-grained forecasting for dense scenarios. The corresponding analysis is provided in Appendix A.5.
>
> **Q3. Have the labels in the 4D dataset been completed to account for occlusions? If they have, can the authors' method handle situations where any frame from the historical RGB images loses an object?**
>
> We visualize the ground truth labels and forecasted occupancy results in Fig. 12 of Appendix A.5.2. It can be observed that:
>
> 1. The occupancy labels of occluded objects remain complete over time, even when an object is nearly missing in some frames;
> 2.  Our method can handle the occluded scenarios to a certain degree, where a historical frame loses an object. Nonetheless, the performance can still be improved in future research.

---

> ### Author Response · Authors · 2024-11-25
> **Official Comment by Authors (Part 2)**
>
> **Q4. Are the 6-DoF camera parameters in the Observer used to align historical 3D features with the current frame? In lines 188-191, why does $F$ change to $F_{motion}$, resulting in an extra matrix dimension of 6×X×Y×Z?**
>
> Yes, the 6-DoF ego-vehicle poses of each frame are used to align historical 3D features into the current frame.
>
> Furthermore, to acquire $F_{motion}$, we perform the following two steps:
> 1. The 6-DoF ego-vehicle poses from $T$ frames are organized into a pose tensor with the shape $T \times 6 \times X \times Y \times Z$, by expanding each 6-DoF pose into 3D and then concatenating them across frames.
> 2. We combine the pose tensor and the 4D feature $F$ (with the shape $T \times C \times X \times Y \times Z$) along channel axis $C$ to produce an ego-vehicle motion-aware feature $F_{motion}$ with the shape $T \times (C+6) \times X \times Y \times Z$.
>
> **Q5. In lines 206-207, how does C+6 become C?**
>
> We apply a $3\times3\times3$ 3D convolution to fuse ego-vehicle poses and environmental information within a local neighborhood, and meanwhile reducing the number of channels from $C+6$ to $C$.
>
> **Q6-(1). What is the difference between the E4A module shown in Figure 4 and the UNet structure?**
>
> There are two differences. From the data format perspective, conventional UNet structures can operate on spatial features like 3D volumes, whereas the E4A module can process spatio-temporal 4D inputs. From the functional perspective, E4A focuses on 4D interaction, by utilizing our Tripling-Attention and the spatio-temporal aggregation.
>
> **Q6-(2). Why does the FLOPs increase significantly while the number of parameters in E4A decreases? Is this due to the upsampling process?**
>
> Yes, this is due to upsampling. The upsampling process generates features with larger sizes, bringing more computations when subsequently operating on these enlarged features. The parameter decrease in E4A benefits from our lightweight design of the Tripling-Attention Fusion (TAF) module.
>
> **Q7. Would cross-attention between different scales yield better results?**
>
> | Method | $\text{IoU}_{c}$ | $\text{IoU}_{f}\ (2.0s)$ | $\tilde{\text{IoU}}_{f}$ |
> |:---:|:---:|:---:|:---:|
> | OccProphet | 34.36 | 26.94 | 29.15 |
> | OccProphet + Cross-Attention | _33.83_ | _26.56_ | _28.77_ |
>
> Thanks for the suggestion. We performed cross-attention between different scales on our OccProphet. However, the performance did not improve, as shown in the table above. A possible reason is that the semantic gap between different scales brings difficulty for learning. How to better design cross-attention for different scales is a valuable research question worth exploring in the future.
>
> **Q8. In the Forecaster, is the condition for prediction learned from past voxels?**
>
> In the Forecaster, the condition is generated from past voxels (relative to future voxels), which consists of previous observations (*i.e.*, frames $1$ to $T-1$) and the current observation (*i.e.*, frame $T$).
>
> **Q9. What does the colon symbol in line 313 mean?**
>
> Sorry for a typo in line 313. We have corrected it with the proper subscript $(T+1:T+T^{\prime},:,:,:,:)$.
>
> The colon symbol in the subscript means collecting features based on the corresponding indices. Specifically, E4A$([O_{obs},F_{future}])$ produces an output in $\mathbb{R}^{(T+T^{\prime}) \times C \times X \times Y \times Z}$. In the subscript, $(T+1:T+T^{\prime})$ means collecting features from the $T+1$ th to $T+T^{\prime}$ th frames. The subsequent colons $(:,:,:,:)$ mean that, for one frame, all features in the voxelized space of that frame are collected. As a result, we obtain a representation $V \in \mathbb{R}^{T^{\prime} \times C \times X \times Y \times Z }$, which is used for subsequent forecasting of occupancy and flow.
>
> **Q10. It would be helpful to follow the output of the 4D BEV method, like [1], with a 2D to 3D detection head for comparison. [1] Bevdet4d: Exploit temporal cues in multi-camera 3D object detection**
>
> | Method | $\text{IoU}_{c}$ | $\text{IoU}_{f}\ (2.0s)$ | $\tilde{\text{IoU}}_{f}$ |
> |:---:|:---:|:---:|:---:|
> | OpenOccupancy-C | 12.17 | 11.45 | 11.74 |
> | SPC | 1.27 | - | - |
> | PowerBEV-3D | 23.08 | 21.25 | 21.86 |
> | BEVDet4D [1] | 31.60 | 24.87 | 26.87 |
> | OCFNet (Cam4DOcc) | 31.30 | 26.82 | 27.98 |
> | **OccProphet (ours)** | **34.36** | **26.94** | **29.15** |
>
> Thanks for the comment. We have adapted BEVDet4D to forecast 4D occupancy for comparison. The experimental results on the nuScenes dataset are reported in the table above. While BEVDet4D achieves considerable forecasting performance, voxel-based approaches such as OCFNet (Cam4DOcc) and OccProphet (Ours) outperform the BEV-based BEVDet4D.

---

> > ### Comment · Reviewer_gfQ3 · 2024-11-26
> >
> > I appreciate the authors' efforts in clarifying the issues in the rebuttal. The rebuttal has addressed all my concerns. Overall it's a ready work for 4D occupancy forecasting. I will increase my score to Accept.

---

> > > ### Author Response · Authors · 2024-11-27
> > > **Official Comment by Authors**
> > >
> > > We are glad to have addressed all your concerns, and deeply appreciate your kind recognition of our work. Thank you sincerely for your great efforts and professional comments in the review!

---

### Meta-Review · Area_Chair_SFCV · 2024-12-20

**Metareview:**

This work proposes a new method for occupancy forecasting in autonomous driving. Its main novelty is a lightweight pipeline outperforming a SOTA approach while incurring a significantly lower computational costs. One of the weaknesses of this work is its clarity that has been raised in different questions by several reviewers. A conceptual shortcoming of the method (as admitted by the authors) is the reduced granularity of forecasting. Increasing it is stated as future research. Overall, the reviewers agree on the merit of this work due to its performance and computational improvements I believe it to be worthy of acceptance.

**Additional Comments On Reviewer Discussion:**

Most of the concerns raised by the reviewers have been addressed by the authors.

---

### Decision · Program_Chairs · 2025-01-22

Accept (Poster)